



# A review of the applicability of the Motivations and Abilities (MOTA) framework for assessing the implementation success of water resources management plans and policies

**John Conallin[1], Nathan Ning[1], Jennifer Bond[1], Nicholas Pawsey[1], Lee J. Baumgartner[1], Dwi Atminarso[1,2,3], Hannah McPherson[4], Wayne Robinson[1], Garry Thorncraft[1,5]**

[1] Institute of Land, Water and Society, Charles Sturt University, PO Box 789, Albury, NSW, 2640, Australia
[2]Research Institute for Inland Fisheries and Fisheries Extensions, Agency for Marine and Fisheries Research, Ministry of Marine Affairs and Fisheries, Jalan H.A. Bastari 08, Jakabaring, Palembang City, South Sumatra 30267, Indonesia.
[3]Inland Fisheries Resources Development and Management Department, Southeast Asia Fisheries Development Center, Jalan H.A. Bastari 08, Jakabaring, Palembang City, South Sumatra 30267, Indonesia.
[4]Hendricks Consulting, Albury, NSW, 2640, Australia
[5]National University of Laos, PO Box 7322, Vientiane, Lao PDR

**Correspondence:** John Conallin (jconallin@csu.edu.au)

**Abstract.** Implementation failure is widely acknowledged as a major impediment to the success of water resource plans and policies, yet there are very few proactive approaches available for analysing potential implementation issues during the planning stage. The Motivations and Abilities (MOTA) framework was established to address this planning stage gap, by offering a multi-stakeholder, multi-level approach to evaluate the implementation feasibility of plans and policies. MOTA is a stepwise process focusing on the relationship between trigger, motivation, and ability. Here we outline the base model of the

MOTA framework and review existing MOTA applications in assorted water resource management contexts. From our review we identify the strengths and limitations of the MOTA framework in various institutional implementation and social adoptability contexts. Our findings indicate that the existing MOTA base model framework has been successful in identifying the motivations and abilities of the stakeholders involved in a range of bottom-up water resource planning contexts, and in subsequently providing insight into the types of capacity- or consent-building strategies needed for effective implementation.

We propose several complementary add-in applications to complement the base model, which specific applications may benefit from. Specifically, the incorporation of formal context and stakeholder analyses during the problem definition stage (Step 1), could provide a more considered basis for designing the latter steps within the MOTA analyses. In addition, the resolution of the MOTA analyses could be enhanced by developing more nuanced scoring approaches, or by adopting empirically proven ones from well-established published models. Through setting the base model application, additional add-in applications can

easily be added to enhance different aspects of the analysis while still maintaining comparability with other MOTA applications. With a robust base model and a suite of add-in applications, there is great potential for the MOTA framework to become a staple tool for optimising implementation success in any water planning and policy-making context.



## 1 Implementation — the Achilles heel of planning and policy success

Implementation failure has long been recognised as a major barrier to planning success (Phi et al., 2015). Policy scientists and planners have persistently attempted to better understand the critical role of implementation in strategic planning and policy making, but often with limited success (Pressman and Wildavsky, 1984; Talen, 1996; Samnakay, 2020). Plan implementation typically involves a variety of stakeholders, who are often required to make significant implementation decisions with regards to how best to translate relatively theoretical plans into plausible outcomes (Joseph et al., 2008). The success of this translation

from planning to implementation, rests largely upon the motivation and abilities of various key stakeholders to see the plan to fruition (Phi et al., 2015).

A range of decision support techniques are available to assess the performance of plans and policies (e.g. Multi-Criteria Analysis (MCA), Cost Benefit Analysis (CBA) (Drèze and Stern, 1987), Robust Decision Making (RDM) and Environmental Impact Assessment (Quan et al., 2019)). These techniques can provide useful information on performance indicators (e.g.

number of houses built, or money saved by protecting against floods etc.). However, performance-based techniques are not sufficient for determining whether a project will actually be adopted by local stakeholders and/or implemented by the appropriate institutions (Quan et al., 2019). The motivations and abilities (MOTA) framework (or MOTA analysis) was developed in 2015 to equip planners and policy-makers with a multi-stakeholder and multi-level approach for assessing the implementation feasibility of projects and plans, centring on the relationship between trigger, motivation and ability (Phi et

al., 2015).

Here we investigate the applicability of the MOTA framework for water resource management. Water resources are critical to supporting food security and energy generation around the globe (IPCC, 2021). Yet, they are becoming increasingly strained in the face of rapid population growth and associated over-development, and this burgeoning pressure will only be exacerbated under changing climatic conditions (IPCC, 2021). Now, more than ever, proficient transdisciplinary approaches

are desperately needed to overcome implementation-blocks to effective and environmentally sustainable water planning and policy success.

The specific objectives of this paper were to (1) review the application of the MOTA framework in various institutional implementation feasibility and social adoptability water resource management contexts, to assess the benefits and limitations of the framework in each context; and (2) use these findings to propose several add-ins to the 'base model' MOTA framework

and identify areas for further research. The review begins by describing the existing base model MOTA framework and its origins, and the process involved in applying the framework. We then provide an overview of the framework's application in institutional implementation and social adoptability water resource management contexts to date, along with benefits and limitations of the framework. The review ends with a description of proposed add-ins to the base model MOTA framework, and recommendations about aspects requiring further research.



## 2 The MOTA framework as a tool for assessing implementation feasibility

The MOTA framework is based on the notion that plan implementation is a multi-actor process (Phi et al., 2015) (Fig. 1). It takes the motivation and ability of actors involved in plan implementation as central and links these to the actors' perceptions of threats and opportunities (Phi et al., 2015). The framework recognises that a particular trigger for an actor may be perceived as a threat, as neutral, or as an opportunity (Quan et al., 2019). Thus, the trigger will influence the actor's level of motivation, which will subsequently influence their action, and ultimately the outcome. A response to a trigger may also be influenced by the actor's ability, and this may consequently influence whether the trigger is perceived as a threat or opportunity (Nguyen et al., 2019a). There is a feedback loop in the MOTA framework, as the outcome resulting from an action can initiate a trigger, leading to a change in perceptions and abilities (Korbee et al., 2019a) (Fig. 1).

In the MOTA framework, abilities are recognized under four categories: financial, technical, institutional and social (Quan et al., 2019) (Fig. 1). Financial ability pertains to having sufficient financial resources to implement the plan (Hoan et al., 2019). By contrast, technical ability collectively refers to the knowledge, skill, expertise, information, tools and materials needed to enact the change  (Hoan et al., 2019). Institutional ability relates to the formal and informal rules that provide a framework for co-ordinating the interactions among groups of actors  (Hoan et al., 2019). It may assist actors in obtaining financial and technical resources from other groups (Phi et al., 2015). Finally, social ability refers to having the competence to effectively interact and communicate with other actors whilst remaining considerate of the key social norms applying to the plan/policy context. It includes aspects such as social cohesion, external networks, trusting the implementing agency, and incorporating inclusive and representative leadership (Sadik et al., 2021).

### 2.1 Origins of the MOTA framework

There are a number of existing stakeholder/actor analysis methods available to water planners and policy makers, covering focus areas ranging from network level (e.g. Configuration analysis (Termeer, 1993)), values (e.g. Analytic Hierarchy Process (Ananda, 2007)), actors' resources (e.g. stakeholder analysis (Bryson, 2004)), and actors' perceptions (e.g. argumentative analysis (Mitroff, 1983)). However, none of them give adequate consideration to behavioural theory (Phi et al., 2015), and many remain fairly abstract and/or qualitative in nature (Bryson, 2004). The MOTA framework provides for a practical stakeholder analysis method incorporating a more comprehensive consideration of behavioural insights (Phi et al., 2015).

MOTA adopts key concepts from three behavioural models: (1) the Theory of Planned Behaviour (TPB) (Ajzen, 1985), (2) the Fogg Behaviour Model (FBM) (2009), and (3) the Motivation-Opportunity-Abilities (MOA) model (Rothschild, 1999) (Fig. 2). The TPB (Ajzen, 1985) proposes that there are three determinants of an intention (i.e. motivation) to perform a behaviour: (1) the attitude towards the behaviour, (2) the subjective norm(s) pertaining to the behaviour, and (3) perceived behavioural control (or perceived ability to perform the behaviour) (Ajzen, 1991). In comparison, the FBM (Fogg, 2009) argues that behaviour is influenced by motivation, ability and triggers. Finally, the MOA (Rothschild, 1999), asserts that actors that are prone, resistant, or unable to respond to a manager's objective behave in accordance with their motivation, opportunity,





and ability. Managers can also use (1) educational strategies to increase the motivation of actors to behave voluntarily (scenario 1 in Fig. 2); (2) marketing to promote alternative opportunities for the actors to behave in line with the manager's objective (scenario 2 in Fig. 2); (3) the law to coerce actors into behaving (scenario's 3 and 4 in Fig. 2); and/or (4) a combination of

educational and marketing strategies to enhance the abilities of the actors to behave (scenario's 5–8 in Fig. 2) (Rothschild, 1999). The MOTA framework draws on the common guiding principle of the TPB, FBM and MOA behavioural models relating to the fundamental importance of motivations and abilities in influencing behaviour (Quan et al., 2019). It then adapts this guiding principle so that the principle can be effectively applied to evaluating the implementation feasibility of plans and policies (Phi et al., 2015).

## 2.2 The process of applying the MOTA framework

The process of applying the MOTA framework can be broken down into six steps (Phi et al., 2015; Quan et al., 2019):

1.      Defining the problem and determining whether MOTA would be applicable

This involves (a) gaining background information on the problem, (b) identifying the relevant stakeholders, (c) defining the spatial and temporal scope of the problem, and (d) refining the final problem definition.

2.      Specifying the relevant MOTA elements

This consists of (a) identifying the current and relevant triggers, (b) defining the expected motivations, and (c) defining the possible financial, institutional, technical, and social abilities.

3.      Preparing the survey(s) to assess the MOTA elements

This involves (a) defining the data collection methods, (b) designing the survey instrument, and (c) pre-testing the survey

(Hoan et al., 2019; Quan et al., 2019).

4.      Implementing the survey(s)

This involves implementing the survey(s) and obtaining an acceptable response rate to reduce the potential for selection biases and/or any other errors.

5.      Processing and analysing the data

This consists of (a) collating, entering and cleaning the survey data, (b) calculating the MOTA scores, (c) mapping the MOTA scores onto a two-dimensional map (with motivation on the x axis and ability on the y axis), and (d) analysing the data (Nguyen et al., 2019b). MOTA scores are calculated by multiplying the average motivation score (-1 to +1) with the average ability score (0 to 1) (Nguyen et al., 2019a).

6.      Synthesising the results and developing recommendations

This involves presenting the results in a way that is useful for planners and decision makers (Quan et al., 2019). This should include providing tangible capacity and/or consent-building recommendations.



## 3 The applicability of MOTA in differing contexts

The MOTA framework was designed to be applicable in differing water management contexts and for various stakeholders (Phi et al., 2015; Quan et al., 2019). Two broad categories of MOTA analysis application are generally recognised: direct plan implementation or governmental implementation feasibility (I-MOTA) and societal adaptation/adoption (A-MOTA) (Quan et al., 2019). I-MOTA involves the government and/or corporate actors who are responsible for facilitating the first and most direct stage of implementation of the official plan, and tends to relate to top-down planning (Phi et al., 2015). A-MOTA involves the actors who are assumed to adapt to the changes prompted by this first stage of plan implementation (i.e. societal actors like citizens, groups of households and communities), and tends to relate to bottom-up planning (Phi et al., 2015).

We reviewed the literature involving applications of MOTA to water resource management, by searching Google, Google Scholar and four databases (CAB Abstracts, ProQuest, ScienceDirect and Web of Science) for the period 2015–2021, using search term combinations from the following terms/phrases: 'MOTA framework', 'Motivations and abilities framework', 'Motivation and Ability (MOTA) framework', 'MOTA analysis', 'Motivations and abilities analysis', 'water', 'adaptive management', 'dissertation', 'thesis', 'Masters', 'Doctoral' and 'conference proceedings' (Table S1, Supplement). Despite the relative newness of the framework (it was first published by Phi et al. (2015)), at the time of writing it has already been applied to 12 studies in two countries (Vietnam and Bangladesh) (Table S1, Supplement). The triggers for MOTA analysis have included climate change impacts (mainly increased flood risk and salinity intrusion) (Nguyen et al., 2019b; Nguyen et al., 2020), the need to modernise the agricultural sector (Korbee et al., 2019b; Korbee et al., 2019a), diminishing groundwater supplies (Pieffers, 2019), and the announcement of participatory water management plans (Sadik et al., 2021). The following section considers the applicability of the MOTA framework in various contexts, by categorising the MOTA literature into I-MOTA studies, A-MOTA studies, and studies involving both I- and A-MOTA.

### 3.1 Governmental implementation feasibility MOTA (I-MOTA)

The applicability of MOTA in top-down planning contexts remains largely underutilised at present and provides opportunities for developing MOTA further in relation to I-MOTA. We found only one example of MOTA being used for exclusively assessing the feasibility of direct plan implementation (Korbee et al., 2019a) (Table S1, Supplement). Korbee et al. (2019a) applied the MOTA framework to evaluate the feasibility of and potential impediments to the implementation of the Mekong Delta Plan in Ben Tre Vietnam, focusing on local- and regional government actors. The authors concluded that the MOTA framework was well-suited to examining government implementation of strategic delta plans; however, they argued that the inclusion of market, civil society and international development actors could provide a more complete assessment of their implementation feasibility and would support the design of a governance framework that could extend beyond the realm of the state (Korbee et al., 2019a).



### 3.2 Societal adoptability MOTA (A-MOTA)

Most MOTA applications to date have been for assessing societal adoption (Table S1, Supplement). The actors of interest in these bottom-up investigations have included farmers, local government staff and other societal actors such as social-based organisations.

#### 3.2.1 Farmers

Five A-MOTA studies have thus far investigated the adaptive capacity of farmers to climate change impacts in the Vietnamese Mekong Delta (Korbee et al., 2018; Hoan et al., 2019; Nguyen et al., 2019b; Nguyen et al., 2020) and Bangladesh (Kulsum, 2020). Korbee et al. (2018), Hoan et al. (2019), Nguyen et al. (2019b) and Nguyen et al. (2020) all examined farmers' behaviours and adaptation intentions to increasing saline intrusion associated with rising sea levels. MOTA was found to be effective for gaining insight into the motivations and abilities of farmers with regard to their adaptation intentions in each study (Korbee et al., 2018; Hoan et al., 2019; Nguyen et al., 2019b; Nguyen et al., 2020), and Hoan et al. (2019) further argued that the framework would be useful for managers and policy-makers in proposing suitable options for carrying out a bottom-up adaptation plan that safeguards the livelihoods of farmers against the effects of saline intrusion (Hoan et al., 2019). Nonetheless, the same authors also suggested that further studies in different contexts (i.e. climate change and climate extremes) and regions were needed to be able to generalise the applicability of the MOTA approach (Hoan et al., 2019).

#### 3.2.2 Local government/other societal actors

The application contexts of MOTA to local government stakeholders and other societal actors have been a little more varied than those pertaining to farmers (Arora, 2018; Korbee et al., 2018; Nguyen et al., 2019a). Nguyen et al. (2019a), for instance, used MOTA to investigate the bottom-up implementation of retrofitting responses to urban flood risk in Ho Chi Minh City (Vietnam), by focusing on District-level Municipality Offices, City-level Sectorial Departments, and Social Mass Organisations (Vietnam Fatherland Front Committee, Vietnam Women's Union, and Ho Chi Minh City Communist Youth Union). The MOTA analysis revealed that the most feasible measure implementable in the short term was a conventional drainage system, as the stakeholders had an average motivation and high ability to implement this type of system (Nguyen et al., 2019a). In comparison, Arora (2018) applied a bottom-up MOTA approach to understand the position of the People's Committee (provincial government) stakeholders with regard to the implementation of Mekong Delta Plan in Ben Tre province (Vietnam), and found the approach to be effective in confirming that officials were positive about the direction of implementation and had no major concerns with adaptation.

#### 3.2.3 Composite contexts involving direct plan implementation and societal adaptation/adoption

Three studies have used MOTA to simultaneously consider both governmental implementation and societal adaptation contexts (Phi et al., 2015; Korbee et al., 2019b; Pieffers, 2019) (Table S1, Supplement). These studies involved various combinations





of stakeholders, including local communities, government agencies, and experts from research institutes and universities in conjunction with officials from government departments and water supply organisations (Phi et al., 2015; Korbee et al., 2019b; Pieffers, 2019). For example, Pieffers (2019) assessed the feasibility of implementing decentralised domestic water provision (DDWP) technologies in the Vietnamese Mekong Delta, to improve water security and reduce the pressure on groundwater supplies in the region. The author used a framework that was based on a qualitative version of MOTA analysis to assess whether the geographical conditions of an area were suitable for DDWP technologies, which asserted that certain conditions needed to be present for an area to be suitable for a DDWP, with those being motivation (social conditions) and abilities (governance, economic, technical and geographic conditions) (Pieffers, 2019). These were categorised as either not present, partially present or present (Pieffers, 2019). By contrast, Phi et al. (2015) investigated the government implementation and social adoptability of strategic planning alternatives for flood management in Ho Chi Minh City. This study was largely intended to be an initial test application of the MOTA framework, and did not attempt to discern between the government and societal actors (Phi et al., 2015). Nonetheless, these distinctions are likely to be significant in strategic planning, since societal actors, like citizens, will most likely play a different role in plan implementation from the government agencies with a formal mandate (Korbee et al., 2019a). Indeed, consumers, citizens and farmers tend to act in uncoordinated ways, but are of great significance as a group because of the significant role they collectively play in the bottom-up implementation of plans. Local and regional government actors, on the other hand, are smaller in numbers, but also play a significant role in the implementation of plans via their institutionalised roles and directives (Korbee et al., 2019a).

## 3.3 Broader uses of the MOTA framework reported in the literature

All of the MOTA studies reviewed argue that the overarching benefit derived from undertaking MOTA analysis is an increased likelihood of achieving plan implementation success (e.g. Phi et al., 2015; Hoan et al., 2019; Korbee et al., 2019a; Nguyen et al., 2019b; Nguyen et al., 2019a). By exploring the interactions between the motivation, ability and trigger components, the MOTA framework can identify potential influencing factors that can be modified so as to achieve a closer match between desired outcomes (those assumed by policymakers and planners) and plausible outcomes (those likely to occur in response to the combined actions of stakeholders during implementation) (Nguyen et al., 2019b). According to Phi et al. (2015), the MOTA framework does this by informing the (re)design of planning elements and planning procedures, to reduce any gaps between planning and implementation. There are two particular areas that can be targeted for process improvement: motivation improvement (consent building) and ability matching (capacity building) (Phi et al., 2015).

### 3.3.1 Motivation improvement (consent building)

Gaining an understanding of motivations using MOTA can be used to identify consent building activities or plan changes. Motivations can be improved by excluding plan options that incite strong opposition, or by altering the plan options (e.g. their spatial or temporal scale) to reduce the negative motivation of certain powerful actors (i.e. those with high ability) (Phi et al., 2015). Alternatively, a potential strategy to appease the less powerful opposed actors (with low MOTA scores) could be to





offer them compensation (Phi et al., 2015) or some sort of incentives (Nguyen et al., 2019b). Nguyen et al. (2019b), for
instance, found that farmers' motivations to adopt new agricultural models in salinity-impacted areas were low, so they
suggested raising motivations by showcasing livelihood models (along with market linkages), providing efficient water
resources, and/or offering agricultural training incentives.

### 3.3.2 Ability improvement (capacity building)

An understanding of the abilities of stakeholders using MOTA, can be used to identify the capacity-building strategies needed
to support a change in behaviour (Quan et al., 2019). Planning options that obtain strong support should be rechecked to ensure
that there is sufficient ability available (Phi et al., 2015). This might involve altering the scope, vision and/or scale of a plan to
allow for the required institutional abilities to be built up. Alternatively, it may involve reallocating or increasing financial and
technical resources to enhance the associated abilities of specific actors that have a significant role in plan implementation (Phi
et al., 2015).

### 3.3.3 Other uses of the framework

In addition to improving the probability of plan implementation success (Phi et al., 2015; Hoan et al., 2019; Korbee et al.,
2019a; Nguyen et al., 2019b; Nguyen et al., 2019a), it was reported that MOTA analysis could be used to help with:

1.      Identifying signals for water resource implementation issues and problems that can be foreseen, earlier in the planning
       process (Phi et al., 2015).
This is likely to save planners and policy-makers much time and money, by equipping them with the capacity to address the
issues and revise the plans at an earlier stage of the planning process.

2.      Determining the underlying trigger factors.
Perceived threats or opportunities behind actors' differing levels of motivations and abilities can be identified using
multivariate analyses, such as Principal Components Analysis (PCA) and/or Hierarchical Cluster Analysis (HCA)  (Nguyen
et al., 2019b).

3.      Examining potential coalitions of actors with (dis)similar motivations and abilities (Nguyen et al., 2019b).
This allows for the segmentation of stakeholders to facilitate more targeted consent- and/or capacity-building strategies.
Analogous segmentation processes have been tried and proven using TPB-based frameworks (Morrison et al., 2012). Morrison
et al. (2012), for instance, used a mixed methodology with a strong TPB-based theoretical foundation to segment landholders
and identify groups, with the overarching goal of improving the targeting of natural resource expenditures.

### 4 Proposed add-ins to the base model MOTA framework

Our review has identified several areas within the MOTA steps where the base model MOTA framework could potentially
benefit from a suite of add-ins to delve deeper into specific areas depending on the application (Fig. 3). These add-ins to the
base model do not take away from the base model application, and their benefit will depend on a range of factors such as depth





of insight and skillsets of the MOTA team in question. Here we describe these add-ins with respect to the steps of the MOTA framework that they apply to.

Our proposed range of add-ins take advantage of the benefits of theoretical triangulation (used in social science) and multiple lines of evidence thinking (used in natural sciences). As summarised by Hoque et al. (2013) this concurrent use of insights from alternative theoretical perspectives minimises the risks associated with attempting to force data to fit a single

theory and the possibility that important insights are unexplored given a theory's failure to cover the issue. In this regard, it has become increasingly common to make use of theoretical triangulation within the acceptance and adoptability literature (Kuehne et al., 2017; Zeweld et al., 2017; Liu et al., 2018; Daxini et al., 2019), and similarly in natural sciences through multiple lines of evidence (Cook et al., 2012).

### 4.1 Incorporating a formal context and stakeholder mapping analyses (Step 1)

The first step of the MOTA framework states the need to identify the problem and relevant stakeholders and define the scope of the investigation within the context of the situation, but offers little further guidance than that. Formal context analyses could be undertaken using several tools during this step to provide a more considered basis for identifying and conceptualising the MOTA elements (i.e. triggers, motivations and abilities) in Step 2 of the framework (Fig. 3). There are numerous formal context analysis tools ranging from simple to complex applications. The minimum should be to gain a good understanding of

the processes within the country the plan is for, and the stakeholders who need to be involved to successfully implement the plan. An example of employing a suite of relatively simple tools is provided below that would provide insight into a country's context. The techniques described would not necessarily need to be conducted by a specialist during the base model MOTA application. To gain an understanding of the country situation, a simple method such as the STEEP (also called VSTEEP or PESTLE) framework could be used (Fleisher and Bensoussan, 2003; Kingsford and Biggs, 2012). STEEP aims to explore the

different values within a system from social, technological, environmental, economic and political (legal) angles, allowing the problem owner to gain an understanding of the country or focus area (Fleisher and Bensoussan, 2003).

The context analysis should also involve gaining an understanding of the different stakeholders using a systematic stakeholder analysis (Reed, 2008; Hermans and Cunningham, 2018). Reed (2008) suggests that the stakeholder analysis process can be broken into three key steps: (1) identifying the stakeholders; (2) differentiating between and categorising the

stakeholders; and (3) investigating relationships between the stakeholders. Of the multitude of stakeholder analysis tools available in the literature (Reed et al., 2009; Bendtsen et al., 2021), tabular forms of stakeholder identification potentially offer the simplest approach to initially listing the different stakeholders and describing how they may affect or be affected by the plan under consideration.

Once groups have been identified, relationships and connections between the different groups can be drawn out using

simple social capital mapping exercises (Conallin et al., 2017). This process allows for the acknowledgement that the various groups identified by the problem owner are not homogenous and different relationships of power, trust and agency may exist within and between groups, as outlined in the previous section. Empirical data collection in the form of key informant



interviews or focus groups could assist in the development of these social capital maps, thereby ultimately improving the conceptualisation of MOTA elements prior to survey design. These social capital maps could be enhanced as part of further

insight analysis in Step 5 (see the section 'Including an explicit further insight analysis') using techniques such as social network analyses to gain a better understanding of the social context (i.e. trust, power, agency).

## 4.2 Visually conceptualising the MOTA elements prior to designing the surveys (Step 2)

Step 2 of the MOTA framework is vital for identifying the relevant MOTA elements and conceptualising the relationships between proposed explanatory and response variables, for designing surveys and interviews (i.e. Step 3). We propose that the

final phase of Step 2 could involve visually conceptualising the potential/hypothesised triggers, motivations and abilities for each actor type being assessed (Fig. 4a), and tabulating the MOTA elements as explanatory and response variables (Fig. 4b) — using a similar approach to that applied in many TPB-based modelling studies (Morrison et al., 2012; Daxini et al., 2019; Rezaei et al., 2019). This would aid in evaluating the contextual logic of the proposed explanatory and response variables in the model, and in turn, ensuring that the MOTA surveys are designed appropriately. It would also allow for the modelling (and

hypothesis testing) of specific relationships between the MOTA elements and/or potential factors influencing the MOTA elements (e.g. Daxini et al., 2019) later on during Step 5 of the framework (Fig. 4b).

## 4.3 Enhancing MOTA scoring precision and the resolution of the results (Step 3)

While the quantitative nature of MOTA allows for more comparisons to be made (see the 'Broader uses' section above), it may provide a false sense of accuracy, with the complexities that characterise implementation processes becoming

oversimplified (Phi et al., 2015). Indeed, the framework's existing mechanisms for quantifying motivations are quite coarse and are well suited to exploratory analyses and making broad comparisons, rather than precise assessments. According to Phi et al. (2015), motivation assessments typically rely on approaches that measure people's preferences. Consequently, they are vulnerable to the problems and limitations known from this field, such as inconsistencies in preference rankings, and the difference between stated and revealed preference. Nonetheless, a useful starting point can involve the adaption of TPB studies

which, through confirmatory factor analysis, have developed survey scales which reliably capture the overall likelihood of an actor undertaking a particular action in the future (e.g. Morrison et al., 2012; Bagheri et al., 2019; Daxini et al., 2019).

Beyond the quantification of actor motivation, ability assessment is not a simple task either. Presently, the MOTA framework relies on a scale from 0 to 1 to express coarse initial assessments of overall abilities (Quan et al., 2019). One approach to better quantify the abilities of societal actors could be to assess them indirectly by taking into consideration their

accessibility to electricity, transport, clean water, livelihood opportunities; personal technical abilities (e.g. education); and roles in social networks (e.g. cultural or religious networks) (Phi et al., 2015). Such data could be collected by undertaking social surveys (Phi et al., 2015). Consistent with the abovementioned refinements to the measurement of motivations, an alternative approach is to leverage prior TPB studies which have considered the role of behavioural controls in constraining the ability of actors to undertake particular actions. In this regard, various TPB studies (Morrison et al., 2012; Liu et al., 2018)



have developed standardised abilities scoring categories with strong potential to enhance the ability analysis component of MOTA. Importantly, TPB based abilities scales use extended Likert scales (i.e. 1 strongly agree to 5 strongly disagree) which provide greater resolution than an ordinal or three-point scales, but are still short enough to enable respondents to discriminate meaningfully between the categories (Hansson et al., 2012).

As mentioned earlier, most of the water resource management-related studies presented in the literature have involved
A-MOTA or a combination of A-MOTA and I-MOTA, whereas only one study, Korbee et al. (2019b), has exclusively involved I-MOTA. Due to the nature of the actors involved in these different MOTA types, the sample size of the populations involved in these studies are likely to vary widely. For example, the population of decision-makers relevant to an I-MOTA study is likely to be relatively small (e.g. 25 local officers interviewed in Quan et al., 2019) compared to the study population within an A-MOTA study (i.e. 50 farmers surveyed in Quan et al., 2019). Consequently, to date, I-MOTA studies have been typically
based on qualitative data (e.g. obtained through interviews or focus groups), whereas A-MOTA studies have been based more on quantitative data generated from questionnaire surveys (Quan et al., 2019). This potentially creates a bias in validity or reliability of the two approaches, highlighting the need for particular attention in the design phase. The incorporation of different steps into the MOTA process, with qualitative data collection prior to the development of the survey or undertaking of interviews may be helpful in minimising any potential biases in I-MOTA studies.

**4.4 Including an explicit 'further insight analysis' (additional detailed analyses for Step 5)**

The existing base model MOTA framework is focused on determining and quantifying the abilities and motivations, and the triggers for these abilities and motivations (Nguyen et al., 2019b). Accordingly, the first five steps described above in the base model MOTA framework will generate critical insights on the triggers, motivations and abilities of different stakeholders needed for the successful implementation of a project. The suggested add-ins will complement the existing base model by
improving the precision in which motivations and abilities can be measured. Knowledge gaps may, however, still exist and the process may benefit from a 'further insight analysis', where specific tools are used to delve into other potential areas of interest that may impact an actor's perceptions towards project opportunities and threats, their overall motivations and the influence of project trigger events. This is consistent with established TPB-based model applications (Morrison et al., 2012; Liu et al., 2018) that are focused more on understanding the factors influencing motivations and the interrelationships between these
factors, and which attempt to untangle the underpinning mechanisms (Borges et al., 2014). A 'further insight analysis' could be useful if wanting to delve further into a particular aspect of a water resources management plan or policy. Below we consider the potential use of Diffusion of Innovations theory, social capital mapping and the enhanced recognition of behaviour factors and social dimensions to support these efforts.

Diffusion of Innovations is a widely used acceptance and adoptability theory with synergistic properties to MOTA.
As outlined by Rogers (2010), Diffusion of Innovations focuses on the processes through which new ideas are communicated through channels amongst members of social systems, and either adopted or rejected. The theory depicts a five-stage innovation-decision process through which decisions makers pass as they overcome uncertainty and gain knowledge of an





innovation, form attitudes towards the innovation, make an adoption or rejection decision, implement the innovation, and ultimately confirm their decisions. A significant body of work has been devoted to understanding the factors that impact the

rate of adoption and Diffusion of Innovations. Amongst others, these factors include the perceived attributes of innovations (i.e. relative advantage, compatibility, complexity, trialability, observability), and type of innovation decision (i.e. optional, collective, authority) (Rogers, 2010). In this regard, Shang et al., (2021) synthesised how perceptions towards the relative advantage, ease of use, compatibility and trialability are often identified as critical factors which impact the adoption of digital technologies within farming systems. These factors are likely to play a key role in helping policy makers to understand actor

how/why different actors are more or less motivated to undertake particular actions, and their perceptions towards potential project opportunities and threats (Fig. 5). Beyond the attributes of the policy intervention, MOTA is also relatively silent in regards to how the individual attributes of actors or behavioural factors (see Burton (2004)) can influence their motivations and attitudes towards an intervention's opportunities and threats. An actor's innovativeness could be a significant consideration given, for instance, the evidence of concerning how some individuals are more willing to be in favour of creative endeavours

and experiment with novel and pioneering approaches (e.g. Pino et al., 2017). Relatedly, an actor's risk preferences or aversion could help to understand why they are more or less motivated given the tendency for some individuals to avoid risks as part of their decision-making processes (e.g. Pannell et al., 2006). Furthermore, and particularly in relation to water policy, interventions which typically involve trade-offs across economic and environmental considerations, environmental concerns and awareness levels are likely to offer significant explanatory power when it comes to understanding actor motivations (e.g.

Läpple and Kelley, 2013) (Fig. 5).

## 5 Areas for further research

The base model MOTA framework was designed to provide a balance between generality/applicability and specificity/accuracy. Our suggested add-ins, without too much additional effort or expertise can improve the predictive and testing capabilities of the base model MOTA framework and allow it to generate more nuanced results for assessing the

implementation feasibility and/or social adoptability of water resource management plans and policies. Nonetheless, there is still much scope to further develop the quantitative capabilities of the MOTA framework, by incorporating new advances from the fields of economics, sociology and psychology; and by conducting further empirical research (Phi et al., 2015). These advances may or may not need to be adapted depending on the objectives of the investigation and context. However, either way, any adaptations or developments to the MOTA framework for these case study applications should be described in

sufficient detail so that it is easily possible to reconstruct their use (Hermans and Thissen, 2009).

Despite the refinements proposed in this review, there are still aspects of the MOTA framework that warrant additional thought. These include further enhancing the quantitative basis of the framework, evaluating its effectiveness in a broader range of water resource management contexts, greater inclusion of social dimensions, and using MOTA as a water resource planning, implementation, and evaluation tool.



MOTA analysis does not consider argumentative analysis, which is part of the shaping of the perceptions among stakeholders (Quan et al., 2019). Similarly, it does not explicitly consider the relationship between actors, and in particular the nature of any alliances or conflicts through techniques such as social network analysis (Quan et al., 2019). Both currently fall outside the scope of the base model MOTA analysis but could be incorporated to allow for greater insight. For example, if a social capital mapping exercise was conducted in Step 1 as part of the context analysis, more in-depth social network analyses
(Bodin et al., 2006; Groce et al., 2019) could be performed to gain further insight into the relationships between different actors. Depending on the initial social capital mapping exercise, network analysis could be used in Step 2 to better design the questions for step's 3 and 4, or as further insight analyses for Step 5.

The MOTA framework currently mentions 'social ability' but doesn't elaborate on it.  Nevertheless, existing frameworks of behaviour, such as Diffusion of Innovations and the Theory of Planned Behaviour, illustrate the critical
dimensions of social relations in individual decision-making, including subjective norms. Diffusions of Innovations, for instance, acknowledges the important role of norms or the "established behaviour patterns for the members of a social system" as potential barriers to change. These norms guide the standards of behaviour of members of a social system and operate at different levels (i.e. organisation, community, nation, local village) (Rogers, 2010). In the TPB, subjective norm refers to the level of perceived social acceptability a particular group gives the potential behaviour (Daxini et al., 2019), or the social
pressure to either perform or not perform a particular behaviour (Ajzen, 1991). These particular groups and associated subjective norms might be linked to key institutions, institutional arrangements or subcultures in natural resource management contexts, where government programs are implemented through intermediary actors and networks (Taylor and Van Grieken, 2015). In the context of natural resource management, particularly for farmers, there may be a perceived responsibility for carrying out a behaviour (McLeod and Hine, 2019), stemming from subjective norms. The current MOTA framework includes
social acceptability as a dimension of motivation (Phi et al., 2015), stemming from Fogg's (2009) behavioural model which also includes a dimension of ability termed 'social deviance', or going against norms. However, we suggest that greater incorporation of social dimensions into the existing MOTA framework could lead to greater depth of analysis and subsequently more fruitful insights into actors' social context and decision-making for water resources management plans and policies.

Social context is a critical consideration in plan implementation since it is key to understanding not only subjective
norms but also behavioural control. The way in which actors are embedded in their dynamic social context, their multi-layered identities, their relationships with others and prevailing power dynamics will shape their agency (Cleaver and De Koning, 2015). Trust between actors, particularly societies' trust in government agencies (i.e. A-MOTA) is influenced by institutional arrangements and will subsequently influence actors' intention to participate in a program (Mettepenningen et al., 2013). For example, farmers are not homogenous groups, instead having varied subcultures or collective norms which influence sets of
values and farming practices and can provide resistance to change (Taylor and Van Grieken, 2015). These subcultures can be influenced by historical and structural factors as well as interactions with peers. Cultural and social capital are therefore important constructs for consideration in understanding different actors' motivations and abilities to participate in water





resources implementation programs (A-MOTA) or make decisions about such programs (I-MOTA). This broader consideration of the social dimensions within MOTA aligns with the multi-step process outlined above.

MOTA has been used as a water resource planning tool in the different contexts described above, but there is also an opportunity to continue to (re)use MOTA throughout the implementation and evaluation phases. The MOTA steps could be revisited and information gained as implementation occurs to evaluate if changes (e.g. motivation, ability) are occurring among the different stakeholders as capacity building and consent buildings programs occur. It is well understood that stakeholders' attitudes, motivations, abilities and relationships change through the implementation process (Sterling et al., 2017). MOTA

has a feedback loop (Fig. 1) that could be fed back into any of the stages to assess these changes and make recommendations on any changes that are needed.

## 6 Conclusion

Current analytical approaches to facilitate and understand strategic planning processes for water resources management typically focus more on the performance of plans rather than the feasibility of plan implementation. The existing base model

MOTA framework attempts to address this void in capacity, by providing a multi-stakeholder and multi-level approach to assess triggers, motivations and abilities underpinning the implementation feasibility of plans. Our review indicates that the existing base model MOTA framework has been effective in determining the motivations and abilities of the stakeholders involved in an assortment of water resources bottom-up planning scenarios, although its mechanisms for quantifying motivations and abilities are quite coarse and probably better suited to exploratory analyses rather than precise assessments.

The base model MOTA framework attempts to find a balance between generality/applicability and specificity/accuracy but is still flexible enough for add-ins when a user wants to delve deeper into a certain aspect of the analysis using other question-specific tools. Running the base model should provide insight into areas that may require further analysis, and there are several add-in applications that can benefit further analyses. We have proposed several add-ins to the existing MOTA framework, which include the incorporation of formal context and stakeholder analyses during the problem definition stage, the

development of more nuanced scoring approaches for undertaking the actual MOTA data collection and analysis stages, and further insight analysis to assess the factors influencing the MOTA elements in addition to the MOTA elements themselves. By eliciting these add-ins and further testing the MOTA framework, it could become a principal approach for achieving planning success in any institutional implementation or social adoptability context — be it water management-related or other.

## Acknowledgments

This study was funded by the Australian Centre for International Agricultural Research (ACIAR) as part of the project FIS/2018/153, 'Translating fish passage research outcomes into policy and legislation across South East Asia'.



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



**Figure captions**


**Figure 1.** The base model MOTA framework (adapted from Phi et al. 2015 and Quan et al. 2019), showing the relationships between the trigger, motivation and ability elements, and the action and outcome. The solid arrows indicate the influence of one element on another, and the dashed arrows indicate a potential influence.

**Figure 2.** The a) Theory of Planned Behaviour model (TPB; adapted from Ajzen 1991), b) Fogg Behaviour Model (FBM; adapted from Fogg 2009), and c) Motivation-Opportunity-Abilities model (MOA; adapted from Rothschild 1999). The TPB proposes that the factors influencing an intention to perform a behaviour comprise of the attitude towards the behaviour, the subjective norm(s) relating to the behaviour, and perceived behavioural control. The FBM argues that behaviour is influenced by motivation, ability, and triggers. The MOA, on the other hand, asserts that motivation, opportunity and ability are principal factors in the performance of an organisation or an individual,

and that strategies inherent in education, marketing and law can be used to enhance these factors where there are inadequacies. The MOTA framework draws out the underpinning concept of these three behavioural models — i.e. the fundamental importance of motivations and abilities in influencing behaviour — and tailors it for assessing the feasibility of planning and policy implementation.

**Figure 3.** An overview of the steps for the base model MOTA framework, and our proposed add-ins to the framework for assessing the

implementation feasibility and/or social adoptability of water resources management plans and policies. TPB = Theory of Planned Behavior.

**Figure 4.** Step 2 of the base model MOTA framework could involve add-ins for (a) visually conceptualising the potential triggers, motivations and abilities for each actor type being assessed, and (b) tabulating the MOTA elements as explanatory and response variables. These add-ins would facilitate verifying the contextual logic of the proposed explanatory and response variables in the model and allow for

the modelling (and hypothesis testing) of specific relationships between the MOTA elements and/or potential factors influencing the MOTA elements. The contents of Figure 4 are for demonstration purposes only, and thus use generically-coded variables. The coded letter(s) used refer to the MOTA element explanatory variables (e.g. '1E' = Hypothesis 1 explanatory variable) and response variables (e.g. 'AR' = Action response variable). In Figure 4b), P values could be hypothesis testing results provided from undertaking regression analysis or another similar type of analysis during Step 5 of the MOTA framework.


**Figure 5.** Step 5 of the base model MOTA framework could involve undertaking 'further insight analysis' to strategically untangle the mechanisms underpinning the MOTA elements for water resources plans. The 'further analysis' add-ins (in the grey shaded boxes) have been adapted from the Diffusion of Innovations (Rogers 2010) and Theory of Planned Behaviour (Ajzen 1991) models.



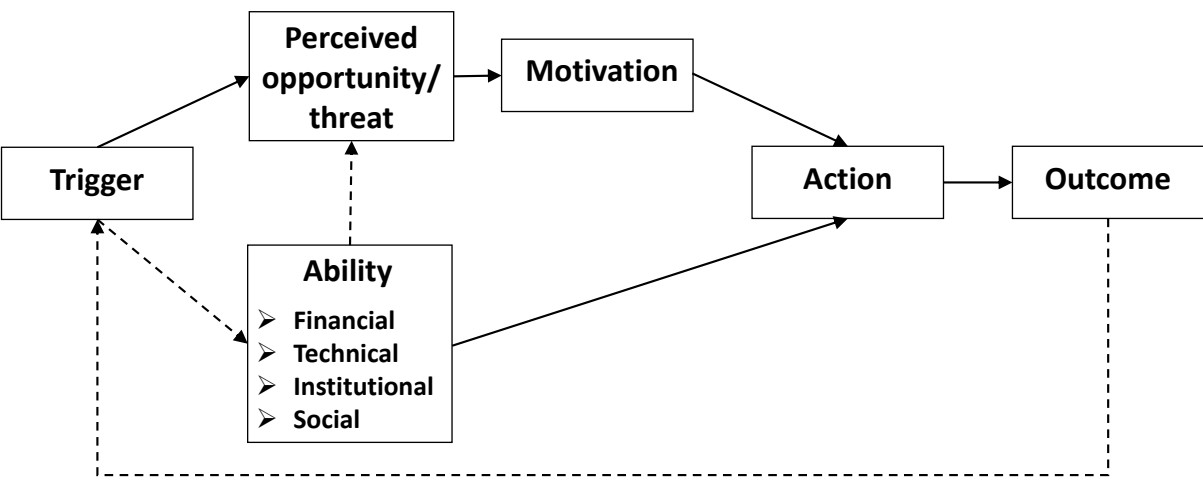

**Figure 1.** The base model MOTA framework (adapted from Phi et al. 2015 and Quan et al. 2019), showing the relationships between the trigger, motivation and ability elements, and the action and outcome. The solid arrows indicate the influence of one element on another, and the dashed arrows indicate a potential influence.





a) Theory of Planned Behaviour model

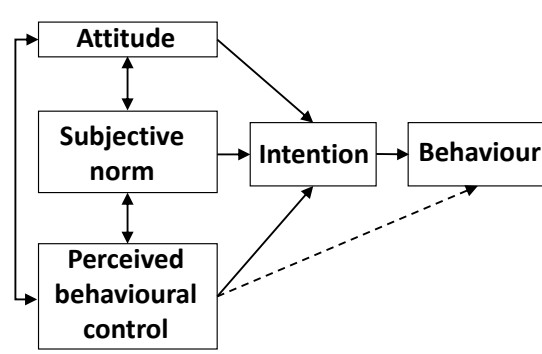

b) Fogg Behaviour Model

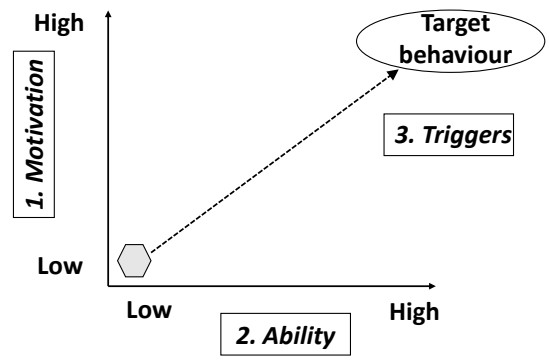

c) Motivation-Opportunity-Abilities model

| Motivation | | Yes | | No | |
|---|---|---|---|---|---|
| Opportunity | | Yes | No | Yes | No |
| Ability | Yes | 1. Prone to behave<br><br>(Education) | 2. Unable to behave<br><br>(Marketing) | 3. Resistant to behave<br><br>(Law) | 4. Resistant to behave<br><br>(Marketing, law) |
| | No | 5. Unable to behave<br><br>(Education, marketing) | 6. Unable to behave<br><br>(Education, marketing) | 7. Resistant to behave<br><br>(Education, marketing, law) | 8. Resistant to behave<br><br>(Education, marketing, law) |

**Figure 2.** The a) Theory of Planned Behaviour model (TPB; adapted from Ajzen 1991), b) Fogg Behaviour Model (FBM; adapted from Fogg 2009), and c) Motivation-Opportunity-Abilities model (MOA; adapted from Rothschild 1999). The TPB proposes that the factors influencing an intention to perform a behaviour comprise of the attitude towards the behaviour, the subjective norm(s) relating to the behaviour, and perceived behavioural control. The FBM argues that behaviour is influenced by motivation, ability, and triggers. The MOA, on the other hand, asserts that motivation, opportunity and ability are principal factors in the performance of an organisation or an individual, and that strategies inherent in education, marketing and law can be used to enhance these factors where there are inadequacies. The MOTA framework draws out the underpinning concept of these three behavioural models — i.e. the fundamental importance of motivations and abilities in influencing behaviour — and tailors it for assessing the feasibility of planning and policy implementation.





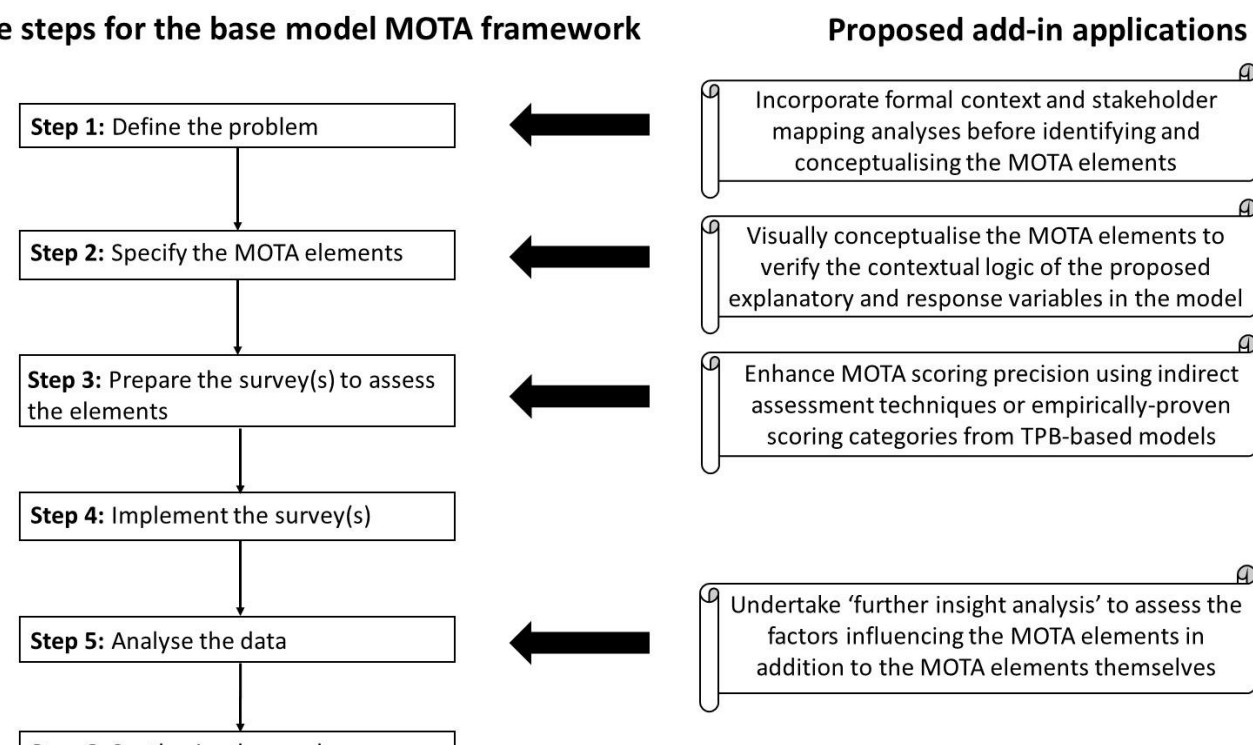


**Figure 3.** An overview of the steps for the base model MOTA framework, and our proposed add-ins to the framework for assessing the implementation feasibility and/or social adoptability of water resources management plans and policies. TPB = Theory of Planned Behavior.





a)

**Stakeholder: Z**

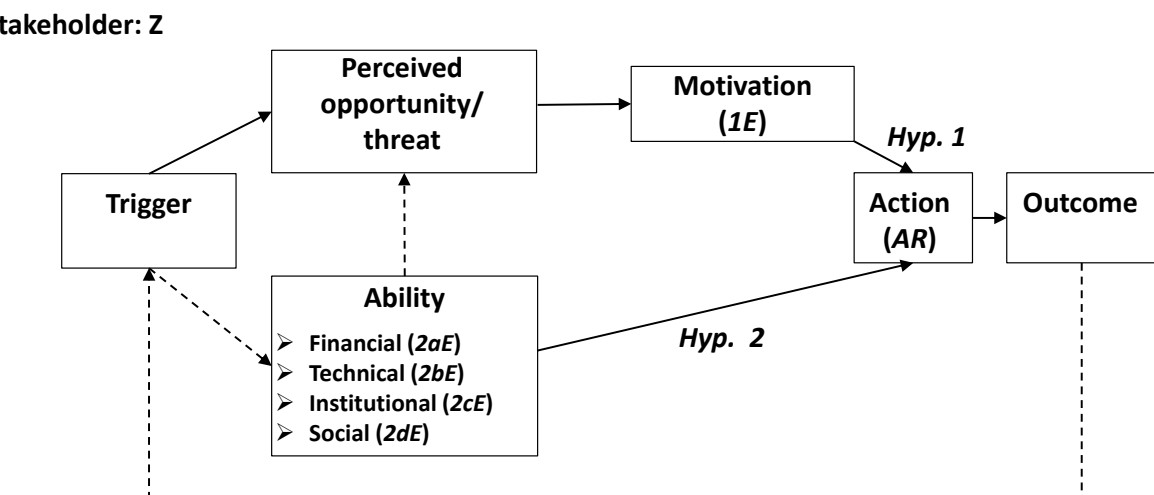

b)

| Hypothesis | Path | *P* value | Result |
|---|---|---|---|
| 1 | Motivation (1E) affects Action (AR) | | |
| 2a | Ability (2aE) affects Action (AR) | | |
| 2b | Ability (2bE) affects Action (AR) | | |
| 2c | Ability (2cE) affects Action (AR) | | |
| 2d | Ability (2dE) affects Action (AR) | | |

**Figure 4.** Step 2 of the base model MOTA framework could involve add-ins for (a) visually conceptualising the potential triggers, motivations and abilities for each actor type being assessed, and (b) tabulating the MOTA elements as explanatory and response variables. These add-ins would facilitate verifying the contextual logic of the proposed explanatory and response variables in the model and allow for the modelling (and hypothesis testing) of specific relationships between the MOTA elements and/or potential factors influencing the MOTA elements. The contents of Figure 4 are for demonstration purposes only, and thus use generically-coded variables. The coded letter(s) used refer to the MOTA element explanatory variables (e.g. '1E' = Hypothesis 1 explanatory variable) and response variables (e.g. 'AR' = Action response variable). In Figure 4b), *P* values could be hypothesis testing results provided from undertaking regression analysis or another similar type of analysis during Step 5 of the MOTA framework.





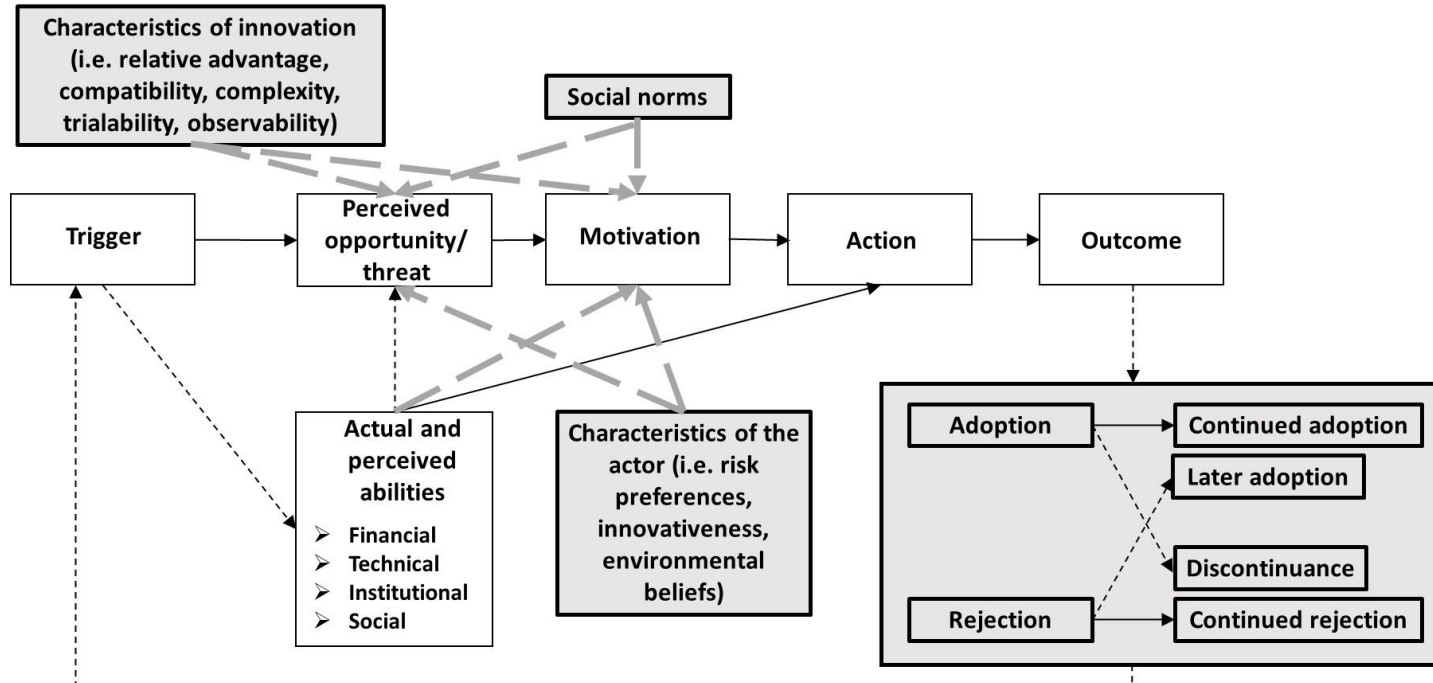


**Figure 5.** Step 5 of the base model MOTA framework could involve undertaking 'further insight analysis' to strategically untangle the mechanisms underpinning the MOTA elements for water resources plans. The 'further analysis' add-ins (in the grey shaded boxes) have been adapted from the Diffusion of Innovations (Rogers 2010) and Theory of Planned Behaviour (Ajzen 1991) models.