# Peer review of "A review of the applicability of the Motivations and Abilities (MOTA) framework for assessing the implementation success of water resources management plans and policies"

_Hydrology and Earth System Sciences, 2021_

## Referee Comment (RC2)

[referee-annotated manuscript omitted]

---

## Author Response (AR1)

**Response to reviewers' comments for initial submission**

Paper: hess-2021-545

**Referee 1**

**General comments**

I think the authors have delivered a good addition to the development of the MOTA framework and they are thus very well contributing to the discussion and development around this framework. I think the paper can be published with minor revisions. Although I do want to give some further thoughts to the authors.

I thank the authors for their interesting contribution to develop and refresh the interest to implementation, as implementation is as stated crucial.

Authors' response and changes

We thank Referee 1 for their constructive comments, and believe that they have made our manuscript much stronger .

**Minor revisions**

1. I advise to get rid of "building" in consent building and especially capacity building. The term building (esp. in development studies) is regarded old-fashioned assuming, you do it from scratch, and if there is no capacity yet. Development, is considered a better term (although also contested for the same arguments), some prefer capacity enhancement/strengthening. Also has a lesser feel of 'social engineering' and that you can top-down change this from external sources.

Authors' response and changes

We have changed the old-fashioned terms of 'consent building' and 'capacity building' to 'consent strengthening' and 'capacity strengthening', respectively.

2. Section 2.1: the authors base this on the work of Phi et al (2015) , but it is not very critical on the body of work in policy analysis that Phi overlooked. For example Contextual Interaction Theory (Bressers, H. T. A. (2004). Implementing sustainable development; how to know what works, where, when and how. In W. M. Lafferty (Ed.), Governance for Sustainable Development: The Challenge of Adapting Form to Function (pp. 248-318). Northhampton, MA: Edward Elgar) and later Owens (Owens, K. A. (2008). Understanding how actors influence policy implementation. A comparative study of wetland restorations in New Jersey, Oregon, The Netherlands and Finland. Enschede: University of Twente.) who applied Contextual interaction theory in quantitative assessments. These also take formal governance systems more in account and power relations between actors.

Authors' response and changes

With respect, we did highlight the need for MOTA to gain further insight into the relationships between different actors, in section 5 (Areas for further research). But we didn't highlight MOTA's gaps concerning policy analysis and formal governance systems.

So we have now highlighted this in our review after considering the references that Referee 1 provided. Thanks for this suggested improvement.

Specifically, in the second paragraph of section 5, we have changed 'These include further enhancing the quantitative basis of the framework, evaluating its effectiveness in a broader range of water resource management contexts, greater inclusion of social dimensions, and using MOTA as a water resource planning, implementation, and evaluation tool.' to:

'These include further enhancing the quantitative basis of the framework, evaluating its effectiveness in a broader range of water resource management contexts, greater consideration of social dimensions, policy analysis and formal governance systems, and using MOTA as a water resource planning, implementation, and evaluation tool.

Also, for the second last paragraph in section 5, we have replaced:

'Social context is a critical consideration in plan implementation since it is key to understanding not only subjective norms but also behavioural control.' with:

'Policy analysis, formal governance systems and social context are all critical considerations in plan implementation (Bressers, 2004; Owens, 2008), but have been largely overlooked by the existing MOTA framework. Social context, in particular, is key to understanding not only subjective norms but also behavioural control.'

The new references have also been added to the reference list accordingly.

**Further thoughts**

1.  It appears that the authors (maybe unknowingly) have positioned themselves in the community that has the view that if we plan more precise (better), are more aware of  that will result in improved implementation (lesser overrun of costs, more timely, less conflict). While our experience in implementation (especially  when implementation happens via projects, which is often in water infrastructure development) is based on incidents, muddling through, very contextual dependent, very experimental (we try this see, we start and find out along the way); implementation is a continued renegotiation of what was planned (on the goals, the resources, allocation and distribution of costs). The MOTA framework does not much include experiential knowledge of implementers and target groups on what works on how to cooperate and renegotiate implementation in the field. It still rationalizes the process of implementation being a logical follow-up of planning, while implementation itself is highly political and a continuation of negotiations and (dis)agreements.

Authors' response and changes
MOTA does have a feedback loop (Fig. 1) that could be fed back into any of the framework's stages to assess how stakeholders' attitudes, motivations, abilities and relationships change through the implementation process, and to make recommendations on any modifications that are needed (see Section 5).

Nevertheless, we completely agree that, in the real world, implementation is a continued renegotiation of what was planned, and that MOTA does not currently offer much scope to incorporate experiential knowledge to effectively implement plans and policies in the field.

To acknowledge this limitation, we have updated the last paragraph of section 5 to:
'MOTA has been used as a water resource planning tool in the different contexts described above, but there is also an opportunity to continue to (re)use MOTA throughout the implementation and evaluation phases. The MOTA steps could be revisited and information gained as implementation occurs to evaluate if changes (e.g. motivation, ability) are occurring among the different stakeholders as capacity strengthening and consent strengthening programs occur. Indeed, in the real world, implementation is a continued renegotiation of what was planned. It is well understood that stakeholders' attitudes, motivations, abilities and relationships change through the implementation process for various reasons (Sterling et al., 2017). MOTA has a feedback loop (Fig. 1) that could be fed back into any of the stages to assess these changes and make recommendations on any modifications that are needed. Nonetheless, there is still much scope to further incorporate the experiential knowledge of implementers and target groups to achieve successful cooperation and renegotiate implementation in the field.'

2. Being more precise in predicting motivations and abilities for a plan can actually result in more problems in implementation. As stakeholder support is very difficult to predict (as it is a heterogenous group) and elements of a plan can be rigid due to legal/contractually binding promises in the plan. As this paper is part of a special issue on transdisciplinary approaches rethinking their own position as rational planners is thus welcomed. The following essay relates to these two camps of planning more precise and planning less and learn more from the practice of implementation: Kreiner, K. (2020). Conflicting notions of a project: The battle between Albert O. Hirschman and Bent Flyvbjerg. Project Management Journal, 51(4), 400-410.

Authors' response and changes

While both Hirschman and Flyvbjerg have contributed immensely to current thinking around project planning and management via their opposing ideas, we argue that the MOTA framework is more in line with Flyvbjerg's philosophy, which aims to "get projects right from the outset" (Flyvbjerg 2017, p. 13). This greater alignment with Flyvbjerg doesn't detract from the acknowledgement that there are uncertainties, unintended consequences and experiential learning within project management.

We have now acknowledged this in section 3.3.3, by changing 'This is likely to save planners and policy-makers much time and money, by equipping them with the capacity to address the issues and revise the plans at an earlier stage of the planning process' to:

'This is likely to save planners and policy-makers much time and money, by equipping them with the capacity to address the issues and revise the plans at an earlier stage of the planning process, with the aim of getting plans or policies "right from the outset" (Flyvbjerg, 2017, p. 13). However, this improved planning phase may still result in unintended consequences in implementation.'

The new reference has also been added to the reference list accordingly.

**General comments**

1. The article beautifully synthesizes the science behind the MOTA framework and its application. The authors also critically review the MOTA framework by bringing more insights and opportunities for further development which would widen the applicability of the framework. The authors also propose three "add-in(s)" for MOTA which would broaden its applicability which is a valuable contribution to the theory as well as to its applicability and flexibility. However, I do have some comments.

Authors' response and changes

We thank the reviewer for their compliments and constructive comments. We have addressed these comments and believe that the manuscript is now much improved.

2. Sections 3 to on-ward found to be mostly based on the literature up to 2019. The recently published literature might have more insights that are potentially missed in this article.

Authors' response and changes

While drafting the paper, we had recently found the Sadik et al. (2020) and (2021) references and added them to the Appendix table, but had forgotten to review them in the main text. We apologise for this oversight.

We have now incorporated reviews of these recent references into the main text by making the following changes:

A. The second paragraph under '3 The applicability of MOTA in differing contexts' now reads:

'Despite the relative newness of the framework (it was first published by Phi et al. (2015)), at the time of writing it has already been applied to 13 studies in two countries (Vietnam and Bangladesh) (Table S1, Supplement). The triggers for MOTA analysis have included climate change impacts (mainly increased flood risk and salinity intrusion) (Nguyen et al., 2019b; Nguyen et al., 2020), the need to modernise the agricultural sector (Korbee et al., 2019b; Korbee et al., 2019a), diminishing groundwater supplies (Pieffers, 2019), and the announcement of participatory water management plans (Sadik et al., 2020; Sadik et al., 2021).'

B. The first paragraph under '3.2 Societal adoptability MOTA (A-MOTA)', now reads:

'Most MOTA applications to date have been for assessing societal adoption (Table S1, Supplement). The actors of interest in these bottom-up investigations have included farmers, local government staff, NGO's and other societal actors such as social-based organisations.'

C. The first paragraph under '3.2.2 Local government/other societal actors', now reads:

The application contexts of MOTA to local government stakeholders and other societal actors have been a little more varied than those pertaining exclusively to farmers (Arora, 2018; Korbee et al., 2018; Nguyen et al., 2019a; Sadik et al., 2020; Sadik et al., 2021). Nguyen et al. (2019a), for instance, used MOTA to investigate the bottom-up implementation of retrofitting responses to urban flood risk in Ho Chi Minh City (Vietnam), by focusing on District-level Municipality Offices, City-level Sectorial Departments, and Social Mass Organisations (Vietnam Fatherland Front Committee, Vietnam Women's Union, and Ho Chi Minh City Communist Youth Union). The MOTA analysis revealed that the most feasible measure implementable in the short term was a conventional

drainage system, as the stakeholders had an average motivation and high ability to implement this type of system (Nguyen et al., 2019a). By contrast, Sadik et al. (2020) used MOTA to assess the implementation feasibility of participatory water management (PWM) reforms in Bangladesh, and found that the framework was capable of informing policymakers and implementing agencies about how to enhance the stakeholders' motivation and ability to ensure effective implementation of PWM reforms. Furthermore, Arora (2018) applied a bottom-up MOTA approach to understand the position of the People's Committee (provincial government) stakeholders with regard to the implementation of Mekong Delta Plan in Ben Tre province (Vietnam), and found the approach to be effective in confirming that officials were positive about the direction of implementation and had no major concerns with adaptation.

D. The first sentence under '3.3 Broader uses of the MOTA framework reported in the literature' now reads:

'All of the MOTA studies reviewed argue that the overarching benefit derived from undertaking MOTA analysis is an increased likelihood of achieving plan implementation success (e.g. Phi et al., 2015; Hoan et al., 2019; Korbee et al., 2019a; Nguyen et al., 2019b; Nguyen et al., 2019a; Sadik et al., 2020; Sadik et al., 2021).'

E. At the end of the paragraph under '4.2 Visually conceptualising the MOTA elements prior to designing the surveys (Step 2)', we have added:

'Sadik et al. (2020) also used causal relationship and indicator mapping to explore the interlinkages among the indicators and relationships among the MOTA elements. This exercise helped them to visualize and understand the MOTA elements, refine the MOTA indicators and improve the survey methodology (Sadik et al., 2020).'

F. We have also added the Sadik et al. (2020) reference into references section and the Table S1, Supplement.

3. The proposed add-in(s) for stage 5 seems to be part of the MOTA components rather than to be standalone components as illustrated in Figure 5, for example:
a. The "characteristics of the actor" add-in is indeed interesting. But how it will be quantified is not clear. Rather than placing it as a standalone component of Motivation, it can be linked with the ability components. For example, the innovativeness characteristics can be included in the technical ability; the environmental belief can be included in the social ability.
b. The "Characteristics of innovation" is interesting. Isn't it already in the Perceived opportunity/threat? For example, if the trigger is an innovation of technology and the action is adoption or rejection of the technology, the MOTA component "perceived O&T" includes advantages of using the technology which might be compatibility, complexity, trialability etc.
c. another proposed add-in is "social norms" which is a part of the institutional ability.

Authors' response and changes

a. As noted below (see response to PDF comment 9), we have amended the Discussion to more clearly acknowledge the potential linkages between abilities (i.e. technical and social) with the characteristics of an innovation and the characteristics of the actor. Figure 5 has also been updated to ensure that these linkages are identified.

b. The characteristics of an innovation is a standalone construct that was adapted from Diffusion of Innovation (DOI). This construct is likely to be a significant factor which will influence an actor's assessment of the perceived O&T associated with an intervention (see response to PDF comment 9 for further details).

c. We have amended the conceptual model (i.e. Fig. 5) to illustrate the relationship between social norms and institutional abilities. As above, the notion of social norms is distinct from institutional abilities and was adapted through the application of Theory of Planned Behaviour (TPB), DOI and other sociological theories.

4.  More explanation on Figure 5 would help the reader. Especially how these add-on(s) can be operationalized and applied could be discussed.

Authors' response and changes

We have provided further suggestions for how various sources of data (i.e. survey, in-depth interviews, archival) can be utilised as part of the measurement of the characteristics of actors and innovations. As part of this processes, researchers can make use of established survey scales and theoretical constructs.

5.  It seems the proposed add-in(s) as illustrated in the Figure 5 are specific to adoption of innovative technology but such explanations are not provided. It is therefore needed to clarify whether these add-in(s) are generalized or specific to the research problem/context of the author.

Authors' response and changes

We have further outlined how, consistent with Rogers (2010), plan implementation can be conceptualised as a form of innovation given the introduction of new ideas.

6.  Further clarification on whether these add-in(s) are for I-MOTA or for A-MOTA would be helpful. Because due the change of actors (from adopting actor to implementing actor) the notion of those add-ins might need to be changed. For example, the Characteristics of Innovations as described in Figure 5 are more relevant to an Implementing Actor (an organization or Agency) rather than an Adopting Actor (Society).

Authors' response and changes

Throughout the paper, we adopt a broad definition of actor to include both individuals (i.e. in the context or A-MOTA) or government/corporate departments/organisations (i.e. in the context of I-MOTA). In line with the reviewer's suggestions, however, throughout section 4.4 (i.e. step 5) we have provided further materials which help to illustrate how the measurement of the constructs might differ between I-MOTA and A-MOTA.

**PDF comments**

1.  Why behavioural theory is important here needs to be addressed first

Authors' response and changes

Yes, this point is relevant. We have added the text: "Yet, the inclusion of behavioural theory is an important consideration due to the active nature of implementation, where stakeholders are required to enact a change"

2.  It seems that the later sections are mostly based on the literature published up to 2020. Although there is a supplementary table listing some reviewed literature in 2021. But those reviews are not found in the relevant sections.

Authors' response and changes

Now addressed. Please see our response and changes made for General comment 2 by Referee 2.

3.  The Kulsum et al 2020 tried to use MOTA to predict the future behaviour of farmers towards a change and tried to interlink MOTA with adaptation pathway, which is an interesting application of MOTA which can be reviewed here. Because, such application of MOTA is a non-traditional use of MOTA.

Authors' response and changes

Thank you for the suggestion. We have further incorporated this into the review section of Appendix A, which contains Kulsum 2020.

4.  Although the MOTA framework does not directly offer any tool for ability improvement, the MOTA indeed offers a systematic approach of assessing the ability towards an action and thus reveals the scopes (opportunities) of ability improvement. Sadik et al 2021 argued that the MOTA framework can be interlinked with other framework for designing ability enhancement/building plans. They showed that extending the MOTA elements up to indicator level helped detailing out the ability. And its livelihood group-wise application could help them further to explore the scopes of designing ability enhancement plan specific to an actor.

Authors' response and changes

The Sadik et al (2021) paper is relevant here, as the indicators allow for greater nuance between ability components to be sought. We have included the following text: 'For example, Sadik et al., (2021) developed a set of indicators for each component of ability, including social ability, thereby further disentangling these elements and allowing for specific areas of ability to be targeted.'

5.  Kulsum et al 2020 showed another use of MOTA. in her research she used MOTA to predict adaptation of action of the farmers in response to the salinity change and uncertainty.

Authors' response and changes

Noted, and this is captured in the text in a more clear and concise way now. In text now - 'Kulsum (2020) used MOTA to predict adaptation pathways of the farmers in response to the salinity change and uncertainty in Bangladesh.'

6.  The another use of MOTA is "Stakeholder Mapping". Besides the original MOTA framework, the MOTA manual (Quan et. al 2019) presents a tool name MOTA Mapping for stakeholder analysis and mapping.

Authors' response and changes

   Yes, and this is a good point as we have discussed in Step 2 creating an initial conceptual MOTA map to aid in a visual representation of where stakeholders are situated in relation to motivations and abilities, and their potential to support or resist a project.

Text has been added to manuscript - 'MOTA mapping, where stakeholders are categorised according to their motivations and abilities, is usually not conducted until Step 5. However, it could be included as part of the initial stakeholder analysis to provide a visual representation of where the problem holder thinks stakeholders sit in relation to motivations and abilities, and their support or resistance to the project. The MOTA map could then be updated as more empirical evidence is collected through step's 2–5.'

7.  This is indeed an important and effective add in. Sadik et al 2020 used a casual-loop/causal relationship and indicator mapping to explore the interlinkages among the indicators and relationship among the MOTA components. Following the Indicator map, the authors further developed a graphical framework to illustrate the relationship among the MOTA components and element. Such exercise helped to visualize and understand the MOTA elements, fine tune the MOTA indicators and improve the overall survey methodology.

Authors' response and changes

Thanks for this information. This has now been incorporated into section 4.2. Please see our changes made in response to General comment 2 by Referee 2.

8.  There is a limitation of statistical application with ordinal number scale. From that perspective, it is better to use cardinal scale/number which would allow further statistical analysis.

Authors' response and changes

Yes, we agree, hence the reason why we have advocated for using extended Likert scales instead of ordinal number scales .

9.  The "characteristics of the actor" add-in is indeed interesting. But how it will be quantified is not clear. Rather than placing it as a standalone component of Motivation, it can be linked with the ability components. For example, the innovativeness characteristics can be included in the technical ability; the environmental belief can be included in the social ability.

Authors' response and changes

This observation makes a good point as there is seemingly some cross-over between the 'add-in' components and the existing constructs within the MOTA framework, such as technical ability which you refer to. However, we think the strength lies in linking the MOTA framework with established theoretical elements rather than trying to fit different behavioural constructs into the existing MOTA. By having these 'add-ins', the process of analysis is more likely to incorporate the vast body of work supporting TPB and DoI (and established mechanisms for measurement of constructs), whereas it might be easier but less effective to incorporate 'bits' of these theories into existing MOTA elements.

We have added the following text: 'Whilst the abovementioned constructs relating to characteristics of an innovation and the characteristics of the actor may seemingly link to different components of MOTA (i.e. actor innovativeness and technical ability, actor environmental beliefs and social ability), we see these characteristics as standalone elements. With connections to established theories (such as TBP or DoI) the additional components allow for more robust and holistic analysis of the implementation action through the body of work underpinning them. This acknowledgement of where the add-ins come from is important to the overall adapted MOTA framework and preferred

over selecting elements of those theories that seem to fit with the existing MOTA framework. Depending on the context and the underlying nature of the actor (i.e. A-MOTA or I-MOTA), as part of the quantification of the characteristics of innovations and of actors, researchers can make use of established theoretical constructs and survey scales. The concepts can be further explored through in-depth, qualitative interviews. Archival data (i.e. financial statements, budgets, strategic documents) may also be particularly relevant for I-MOTA analysis given the insights they may reveal about an organisation's size, performance, willingness to invest in innovation, and environmental credentials.'